# FROM GENERATION TO RESTORATION IN SINGLE-IMAGE REFLECTION REMOVAL

## ABSTRACT

Single-image reflection removal (SIRR) is a highly ill-posed problem, where existing discriminative methods struggle to recover regions heavily corrupted by reflections and often fail to generalize in the wild. This work presents a new framework that reframes SIRR as a guided generation task by adapting a pre-trained Diffusion Transformer (DiT) into a precise restoration model. The key principle is to regulate the generative flexibility of DiTs within a structured latent space. To this end, we design two core components, including i) *a reflection-equivariant VAE* that encodes reflection artifacts into a compact latent prior; and ii) *a set of learnable prompts* that provides direct, task-specific guidance while bypassing the ambiguity of text-based conditioning; These designs transform a general-purpose image editing DiT into a precise and robust tool for reflection removal, capable of reconstructing transmission layers with high fidelity and fine detail. Extensive experiments reveal that our model achieves new state-of-the-art performance on standard benchmarks and, critically, generalizes strongly to challenging real-world images. Code will be made publicly available.

## 1 INTRODUCTION

*What I cannot create, I do not understand.* – Richard Feynman

Capturing photos through (semi-)transparent surfaces like windows is a common practice in daily life, yet it often produces images interfered by reflections. These reflections superimpose distracting patterns onto the transmission layer, undermining both the aesthetic quality of the photograph and the reliability of downstream vision tasks, such as depth estimation, stereo matching, and optical flow (Tsin et al., 2003; Yang et al., 2016; Costanzino et al., 2023; Jiang et al., 2024), to name just a few. Single-image reflection removal (SIRR) addresses this challenge by decomposing a composite image into its transmission and reflection layers. The task is fundamentally ill-posed, since a single observed image admits infinitely many possible decompositions, making it a longstanding issue in computational photography and image restoration.

Early approaches tackled SIRR using optimization with handcrafted priors, such as sparsity (Guo et al., 2014) or ghosting cues (Shih et al., 2015). While conceptually elegant, these methods relied on strong assumptions that rarely hold in complex, real-world scenes. With the rise of deep learning, CNN-based methods trained on large-scale synthetic datasets (Zhang et al., 2018; Wei et al., 2019) brought significant progress. Pioneering works leveraged perceptual features from pre-trained networks like VGGNet to separate transmission and reflection layers (Li et al., 2020; Hu & Guo, 2021), demonstrating the effectiveness of semantic priors. Subsequent models (Hu & Guo, 2023; Zhao et al., 2025; Hu et al., 2024) advanced this line by incorporating stronger backbones, such as Swin Transformers (Liu et al., 2021) and Focal Networks (Yang et al., 2022), to better preserve and propagate discriminative clues. Despite these advances, existing methods face two critical limitations. First, their inherently discriminative nature restricts them to pixel-to-pixel mappings, making it difficult to plausibly reconstruct scene content where the transmission layer is heavily occluded. Without a generative prior, they often produce residual reflections or overly smoothed outputs. Second, they show limited generalization, struggling with diverse, in-the-wild inputs that deviate from synthetic training distributions. One may refer to Fig. 1 for illustrative examples.

The recent revolution in generative modeling, fueled by the scalability of Diffusion Transformers (DiTs) (Peebles & Xie, 2023) trained on web-scale data, has redefined state-of-the-art image syn-

Figure 1: Examples of two types of challenges. In comparison with the previous SOTA method DAI (Hu et al., 2025), our model exhibits clear advantages in these challenging scenarios.

thesis. Models such as Stable Diffusion 3 (Esser et al., 2024) and FLUX.1 (Black Forest Labs, 2025) showcase breathtaking photorealism and increasingly controllable image editing, highlighting the immense potential of DiTs as powerful world priors.

Yet this generative capacity is a double-edged sword for meticulous restoration tasks like SIRR. On the one hand, DiTs can plausibly reconstruct heavily occluded regions, where discriminative models typically fail. On the other hand, their creative freedom can easily become a liability, say naively prompting a model to "remove the reflection" may yield incorrect textures, distort geometry, or overwrite details that should be preserved. Unlike open-ended generation, SIRR demands faithful disentanglement of the transmission and reflection layers. This tension exposes the central dilemma, *i.e., how can we harness the generative strength of DiTs while constraining them to act as reliable restoration tools?* We argue that the solution lies in taming the generative process through principled mechanisms, which raises two key questions: 1) how to establish a latent space that faithfully represents the reflection, transmission, and their mixture, and 2) how to instruct the model to execute this highly specific removal task without relying on ambiguous natural language.

To address these aforementioned challenges, this work develops a novel generative framework with the primary contributions summarized as follows:

- We present the first framework to repurpose a large-scale Diffusion-based image editing model for single-image reflection removal, providing a principled blueprint for adapting large-scale generative models to restoration tasks.

- To ensure faithful reflection removal, we design a reflection-equivariant variational autoencoder (VAE) that encodes a compact latent prior representing the contaminating input and employ learnable prompts as direct, optimized embeddings to instruct the model.

- Extensive experiments are conducted to demonstrate that the approach achieves state-of-the-art results on standard benchmarks and delivers superior visual quality and robustness on challenging real-world images, addressing limitations of prior methods.

## 2 RELATED WORK

### 2.1 SINGLE IMAGE REFLECTION REMOVAL

**Prior-based Methods.** Pioneering work on SIRR formulated the task as an optimization problem, regularized by handcrafted priors derived from the physical properties of the statistical regularities of natural images (Levin & Weiss, 2007). These methods sought the most plausible decomposition by minimizing an objective function that balanced data fidelity with these priors. Prominent

examples include relative smoothness assumptions (Chung et al., 2009; Li & Brown, 2014), where the transmission layer is expected to be smoother than the reflection, gradient sparsity (Levin et al., 2002; 2004; Fan et al., 2017), and the detection of ghosting cues from double reflections (Shih et al., 2015). While insightful, the efficacy of these methods is fundamentally constrained by a reliance on fragile assumptions, which are frequently violated in complex, real-world scenes. Consequently, the performance is often limited to controlled environments, exhibiting poor generalization to in-the-wild data (Wan et al., 2018). Nevertheless, these prior-based approaches established the conceptual groundwork for the field, influencing the design of subsequent deep-learning methods.

**Learning-based Methods.** The advent of deep learning (Simonyan & Zisserman, 2015; He et al., 2016), particularly Convolutional Neural Networks (CNNs), marked a paradigm shift in SIRR (Zhang et al., 2018). By training on synthetic datasets, these methods learn to perform the decomposition in an end-to-end fashion. The foundational insight was the use of semantic priors from pre-trained classification networks. Zhang *et al.* (Zhang et al., 2018), for instance, pioneered this by leveraging hypercolumn features from VGG-19 (Hariharan et al., 2015) to imbue their model with greater semantic awareness, while ERRNet (Wei et al., 2019) further explored this by training with misaligned image pairs. This core concept spurred a wave of architectural innovation, with increasingly sophisticated modeling of the relationship between the transmission and reflection layers. One strategy is a two-stage approach, where the network first estimates one component to guide the prediction of the other. RAGNet (Li et al., 2023) initially estimates the reflection to guide transmission recovery. A parallel line of inquiry pursues simultaneous estimation through dual-stream networks. The YTMT strategy (Hu & Guo, 2021) exemplifies this by restoring both layers concurrently with an interactive module; however, its reliance on a linear physical assumption limited its performance. Other works, such as BDN (Yang et al., 2018) and IBCLN (Li et al., 2020), employ iterative refinement, but their simpler interaction models can sometimes lead to heavy ghosting artifacts. More recent architectures have introduced even more complex interaction mechanisms. Dong *et al.* (Dong et al., 2021) developed an iterative network that estimates a probabilistic reflection confidence map, while DSRNet (Hu & Guo, 2023) introduced a mutually gated interaction mechanism. To capture long-range dependencies, Transformer-based architectures like the DSIT (Hu et al., 2024) and RDNet (Zhao et al., 2025) have also been adapted, pushing the state-of-the-art on benchmark datasets. Recently, RRW (Zhu et al., 2024) proposed to detect the reflection area and then execute removal. Despite their success, these methods are fundamentally discriminative;*i.e.* a deterministic pixel-to-pixel mapping. This leads to two critical limitations: (1) a failure to reconstruct content in regions heavily occluded by reflections, often resulting in blurriness or artifacts, and (2) poor generalization to in-the-wild images whose statistics differ from the training data.

**Generative Methods.** To overcome the limitations of discriminative approaches, some works have turned to generative modeling. Early explorations in this domain utilized Generative Adversarial Networks (GANs) (Fan et al., 2017; Hu & Guo, 2021; 2023) to enhance the perceptual quality of the restored transmission layer. While GANs can produce sharp textures, their adversarial training is notoriously unstable, and they often struggle with new artifacts. The recent ascendancy of diffusion models has opened a more promising avenue, given their demonstrated power in general image restoration tasks like super-resolution (Fei et al., 2023) and inpainting (Rombach et al., 2022). However, adapting these models for SIRR is non-trivial, as the task requires precise layer disentanglement rather than detail synthesis. Initial explorations into diffusion-based SIRR, such as L-DiffER (Hong et al., 2024) and PromptRR (Wang et al., 2024), have leveraged language guidance as an additional conditioning signal. This approach, however, faces a critical bottleneck: the ambiguity and difficulty of describing complex, non-semantic reflection patterns in text. Furthermore, these models are typically trained from scratch on task-specific datasets, limiting their access to the vast world knowledge embedded in large-scale, pre-trained foundation models. Another recent work, DAI (Hu et al., 2025) attempts to leverage a one-step diffusion prior. However, this method fundamentally operates as a discriminative, pixel-to-pixel regression network, using the diffusion model as a prior rather than a generative engine. As such, it sacrifices the very generative capabilities needed to reconstruct heavily occluded regions (see Fig. 1 for a visual comparison). How to harness the immense generative power of a *pre-trained* foundation model for SIRR without resorting to imprecise textual control or reverting to a discriminative framework remains a problem.

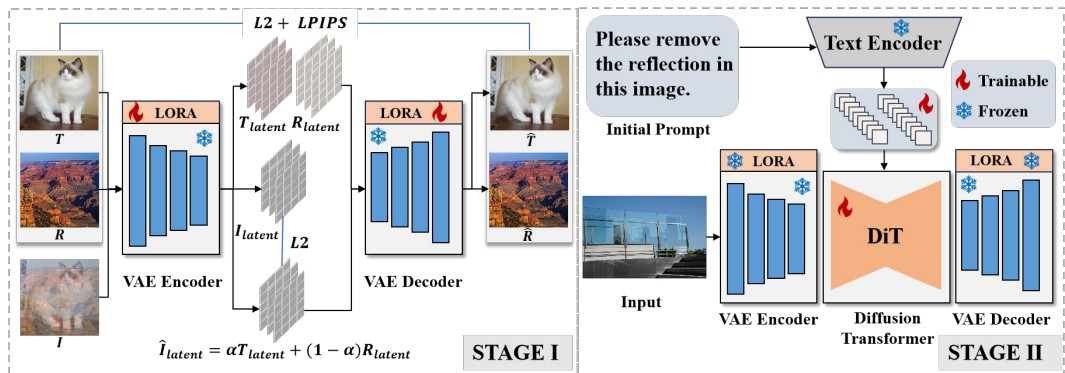

Figure 2: The overall pipeline of our proposed method. In Stage I, we use reflection-equivalence loss to regularize the latent space; during Stage II, the VAE encoder and decoder are frozen, with a learnable text prompt and the DiT model to be trained;

## 2.2 GENERATIVE IMAGE EDITING

Our work is situated within the rapidly advancing field of generative image editing. While early methods often involved manipulating the latent space of GANs (Ling et al., 2021; Wang et al., 2022; Pan et al., 2023), the current state-of-the-art is now defined by large-scale text-to-image diffusion models like Stable Diffusion (Rombach et al., 2022), Imagen (Saharia et al., 2022), and FLUX.1. The evolution of control mechanisms for these models has followed several key paradigms. The first and most common mode of control is the text prompt. A significant body of work, such as InstructPix2Pix (Brooks et al., 2023), has focused on training models to follow editing instructions in natural language. While versatile for creative tasks, text is ill-suited for restoration tasks like SIRR, as describing the precise instructions of removing a reflection in words is both impractical and ambiguous. A second paradigm introduced explicit structural conditioning. Landmark methods like ControlNet (Zhang et al., 2023) and T2I-Adapter (Mou et al., 2024) allow users to guide generation with auxiliary inputs like edge maps or depth maps. Works like OminiControl (Tan et al., 2024) and UNO (Wu et al., 2025b) have explored fine-grained control by manipulating the model's internal features. While these provide powerful, general-purpose toolkits for manipulation, they often struggle with the high-fidelity content preservation required for restoration. Most recently, the field has seen the rise of massive, general-purpose image editing models like GPT-4o, FLUX.1 Kontext (Black Forest Labs, 2025), and Qwen3-Image-edit (Wu et al., 2025a), which have demonstrated remarkable progress in instruction-following and content-preserving editing. However, their output fidelity, while impressive, often falls short of the stringent requirements for high-precision image restoration tasks (Yang et al., 2025a). Furthermore, their reliance on the aforementioned control paradigms still limits their applicability to tasks like SIRR. In this work, we leverage these powerful foundation models as a starting point and introduce the crucial missing piece for adapting them to the specialist task of SIRR.

## 3 METHODOLOGY

Our method is designed to tame a powerful pre-trained image editing DiT [1] for the precise and robust task of removing reflections from a single image. To realize this, our methodology is composed of two components, as illustrated in Figure 2: (1) a reflection-equivariant VAE that represents the reflection, transmission, and their mixture in a faithful manner, and (2) a set of learnable prompts that instruct the DiT to perform the removal task.

## 3.1 RECTIFYING THE LATENT SPACE FOR REFLECTION REMOVAL

Our investigation begins with the VAE component of the pre-trained FLUX.1 model, which is responsible for encoding the reflection-corrupted input image. As shown in Table 1, while the original VAE excels at reconstructing standard images, its performance (measured by SSIM) degrades significantly on synthetic mixtures of background and reflection layers. Given encoder $E$ and decoder $D$,

---

[1]Here, we choose FLUX.1 Kontext (Black Forest Labs, 2025) as our base model.

Table 1: SSIM comparison on synthetic VAE reconstruction and real-world SIRR. While a simple reconstruction finetune helps reconstruct synthetic mixtures, it fails in the SIRR task. Our method succeeds on both, demonstrating the importance of a properly structured latent space.

| Model | Non-Mixed Recon. | 50% Mixed Recon. | SIRR on Real20 |
|---|---|---|---|
| Vanilla FLUX-VAE | 0.909 | 0.859 | 0.841 |
| Fine-tuning with $\mathcal{L}_{\text{recon}}$ | 0.918 | 0.876 | 0.842 |
| Ours($\mathcal{L}_{\text{recon}} + \mathcal{L}_{\text{equiv}}$) | **0.919** | **0.877** | **0.871** |

a straightforward solution is to finetune the VAE on these mixed images using a standard pixel-wise reconstruction loss, as suggested by stable diffusion (Rombach et al., 2022):

$$\mathcal{L}_{\text{recon}} = \|D(E(x)) - x\| + \text{LPIPS}(D(E(x)), x). \tag{1}$$

While this baseline approach improves the reconstruction metric on synthetic data by brute-forcing the VAE to accommodate the new distribution, the lack of structure proves critical when moving from reconstruction to reflection removal. Our goal is therefore not merely to improve reconstruction metrics, but to restructure the latent space to be *reflection-equivariant*. We aim to align the VAE's latent geometry with the linear physics of reflection formation, where an observed image $I_{\text{obs}}$ can be seen as a linear blend of a background $B$ and a reflection $R$: $I_{\text{obs}} \approx (1 - \alpha)B + \alpha R$. We enforce this property by finetuning the encoder, $E$, such that the encoding of the mixture is approximately equal to the mixture of the encodings:

$$E(I_{\text{obs}}) \approx (1 - \alpha)E(B) + \alpha E(R). \tag{2}$$

To achieve this, we introduce a crucial *equivariance loss*, $\mathcal{L}_{\text{equiv}}$, alongside the standard reconstruction loss. During training, we sample a background $B$, a reflection $R$, and a random interpolation factor $\alpha \in [0, 1]$ to create the mixture $I_{\text{obs}}$. The equivariance loss then penalizes any deviation from the desired linear behavior in the latent space:

$$\mathcal{L}_{\text{equiv}} = \|E(I_{\text{obs}}) - ((1 - \alpha)E(B) + \alpha E(R))\|_1. \tag{3}$$

As shown in Table 1, our method achieves reconstruction scores on synthetic data comparable to the baseline. The true benefit of our approach, however, is demonstrated in the SIRR performance. By explicitly enforcing linearity, our finetuned VAE learns a smooth, well-behaved space that generalizes beyond the synthetic training data. This proper latent structure is the key to its superior performance on real images, where the unstructured MSE-finetuned VAE fails to perform.

### 3.2 LEARNABLE PROMPTS FOR PRECISE TASK GUIDANCE

A significant challenge in adapting pre-trained image editors for a restoration task like SIRR is bridging the semantic gap between human instruction and the model's operational capabilities. Text prompts, the standard modality, are fundamentally ill-suited for this task. It is infeasible to describe the complex, non-semantic patterns of an arbitrary reflection in words, and a generic prompt like remove the reflection is too ambiguous and doesn't work in practice. To overcome this, we introduce a set of *learnable prompts*, denoted as $P_{\text{task}}$, which are optimized to become a highly specialized, non-verbal task embedding. Our approach connects the model's text-based pre-training with our task-specific fine-tuning through a strategic initialization and optimization process.

**Semantic Initialization.** We begin by initializing the prompt by tokenizing a simple, descriptive sentence: "please remove the reflection within the image." The resulting sequence of text embeddings serves as initialization for our learnable prompt vectors. This initialization is crucial; it leverages the model's semantic pre-training, placing the optimization of the task prompt in a semantically meaningful starting point of the embedding space. The model already possesses an understanding of concepts, providing a strong inductive bias and a good starting point for optimization.

**Optimization into a Task Vector.** These initialized vectors are replacing the output of text encoders. During fine-tuning, the prompt vectors are untethered from their linguistic origins and are optimized via backpropagation. This allows $P_{\text{task}}$ to evolve from a generic, human-readable instruction into a direct, optimized task embedding that exists in the model's native operational space. In essence, this

Table 2: Quantitative comparison with SoTA methods on public SIRR benchmarks (Real20, SIR2, Nature) and the average scores. Best results are in **bold** and second best are underlined.

| | Methods | Real20 (20) | | SIR2 (454) | | Nature (20) | | Avg. | |
|---|---|---|---|---|---|---|---|---|---|
| | | PSNR | SSIM | PSNR | SSIM | PSNR | SSIM | PSNR | SSIM |
| Non-Gen. | ERRNet(CVPR'19) | 22.89 | 0.803 | 23.55 | 0.882 | 22.18 | 0.756 | 23.47 | 0.874 |
| | IBCLN(CVPR'20) | 21.86 | 0.762 | 24.20 | 0.884 | 23.57 | 0.783 | 24.08 | 0.875 |
| | YTMT(NeurIPS'21) | 23.26 | 0.806 | 24.08 | 0.890 | 23.85 | 0.810 | 24.04 | 0.883 |
| | Dong *et al.*(ICCV'21) | 23.34 | 0.812 | 24.25 | 0.901 | 23.45 | 0.808 | 24.18 | 0.894 |
| | DSRNet(ICCV'23) | 23.91 | 0.818 | 25.71 | 0.906 | 25.22 | 0.832 | 25.62 | 0.899 |
| | Zhu *et al.*(CVPR'24) | 21.83 | 0.801 | 25.48 | 0.897 | 26.04 | 0.846 | 25.37 | 0.909 |
| | DSIT(NeurIPS'24) | 25.22 | 0.836 | 26.43 | 0.911 | 26.77 | **0.847** | 26.40 | 0.905 |
| | RDNet(CVPR'25) | 25.71 | 0.850 | 26.69 | 0.908 | 26.31 | 0.846 | 26.63 | 0.903 |
| Gen. | DAI (ArXiv'25) | 25.21 | 0.841 | 27.47 | 0.919 | 26.81 | 0.843 | 27.35 | 0.913 |
| | Ours | **27.27** | **0.871** | **27.99** | **0.921** | **27.30** | 0.838 | **27.93** | **0.916** |

Table 3: Comparison of state-of-the-art methods on the OpenRR benchmark. Please note that none of these methods, including ours, are trained on the OpenRR training set. Best results are in **bold**.

| Metric | ERRNet | IBCLN | YTMT | Dong *et al.* | DSRNet | DSIT | RDNet | DAI | Ours |
|---|---|---|---|---|---|---|---|---|---|
| PSNR | 22.60 | 24.33 | 22.20 | 23.76 | 23.30 | 24.77 | 24.87 | 25.27 | **27.76** |
| SSIM | 0.802 | 0.930 | 0.797 | 0.817 | 0.803 | **0.869** | 0.850 | 0.831 | 0.843 |

process distills the abstract concept of reflection removal into a set of precise, low-level instructions that are directly interpretable by the DiT's attention mechanisms. The learnable prompt effectively becomes the operational command that instructs the DiT on *how* to utilize the information provided by the input. This provides a clear and highly effective way to guide a large-scale generative model.

## 4 EXPERIMENTS

### 4.1 IMPLEMENTATION DETAILS

Our entire framework is implemented in PyTorch and consists of a two-stage training process: first training the reflection-equivariant VAE, and then fine-tuning the Diffusion Transformer.

**Reflection-Equivariant VAE Training.** The first stage focuses on training our VAE to learn a structured latent space for reflection. To make sure that the latent space won't change dramatically, we train a Low-Rank Adaptation (LoRA) adapter with a rank of 8. The training is conducted for 30,000 iterations using the AdamW optimizer with a learning rate of 1e-4. To learn a robust and general representation of reflections, we use the high-quality PD-12M (Meyer et al., 2024) dataset, exposing the model to approximately 3.84 million unique images with a global batch size of 128.

**DiT Fine-Tuning.** The second stage involves fine-tuning the main reflection removal model. We initialize our model from a pre-trained FLUX.1 Kontext checkpoint to leverage its powerful generative prior and extensive world knowledge. The full model is then fine-tuned using the AdamW optimizer with a fixed learning rate of 1e-5 and a batch size of 32. Following established best practices in the field, our training data is a curated mixture of real-world and synthetic image pairs. To ensure high-fidelity data augmentation, the synthetic pairs (composite and transmission) are generated using the physically-grounded pipeline and model proposed by RDNet (Zhao et al., 2025).

### 4.2 QUANTITATIVE PERFORMANCE EVALUATION

**Datasets.** Our evaluation is performed on several widely-used benchmark datasets. For quantitative analysis, we use four standard test sets: *Nature* (Dong et al., 2021), *SIR2* (Wan et al., 2022), *Real20* (Zhang et al., 2018), and *OpenRR* (Yang et al., 2025b).

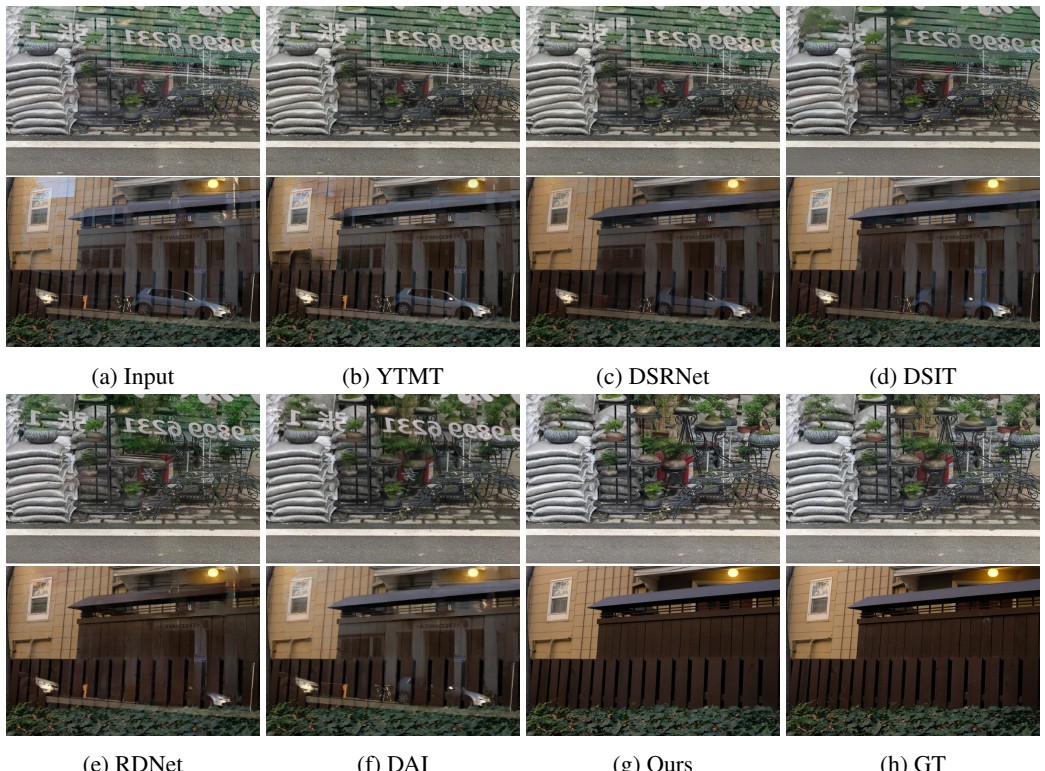

| (a) Input | (b) YTMT | (c) DSRNet | (d) DSIT |
|---|---|---|---|

| (e) RDNet | (f) DAI | (g) Ours | (h) GT |
|---|---|---|---|

Figure 3: Qualitative comparisons on Real20 and OpenRR. Please zoom in for more details.

**Evaluation Metrics.** We use the two most common metrics in image restoration to measure the quality of the recovered transmission layer: Peak Signal-to-Noise Ratio (PSNR) and Structural Similarity Index Measure (SSIM). For both metrics, higher values indicate better performance.

**Baselines.** We compare GenSIRR against a representative set of recent and influential SIRR methods, including the CNN-based ERRNet (Wen et al., 2019), IBCLN (Li et al., 2020), YTMT (Hu & Guo, 2021) and Dong *et al.* (Dong et al., 2021) as well as more recent architectures like DSIT (Hu et al., 2024) and RDNet (Zhao et al., 2025). We also included DAI (Hu et al., 2025), a model trained with a more sophisticated dataset and large-scale pretraining. These baselines cover a range of architectural designs and reflect the current state-of-the-art.

**Results on Publicly Available Datasets.** As shown in Table 2 and Table 3, GenSIRR establishes a new state-of-the-art across all benchmark datasets. Our method consistently outperforms all baseline models in both PSNR and SSIM, often by a significant margin. The performance gains are particularly pronounced on the more challenging datasets like Real20 and OpenRR, which contain complex structures and diverse reflection types. This demonstrates the superior capability of our generative approach to handle difficult cases where previous discriminative models struggle. The strong results on the real-world SIR2 dataset underscore the excellent generalization ability of our framework, a key weakness we aimed to address.

### 4.3 QUANTITATIVE PERFORMANCE EVALUATION

**Visual Results.** We provide visual comparisons in Figure 3 to highlight the qualitative superiority of GenSIRR. On challenging examples, our method is visibly more effective at removing complex reflections while preserving scene fidelity compared to prior state-of-the-art approaches.

**Generalization to Challenge In-the-Wild Images.** The true strength of GenSIRR is most evident in its performance on real-world images, as shown in Figure 4. While competing methods often fail to adapt to the domain shift from synthetic training data, producing unnatural artifacts or incomplete removal of reflections, our model demonstrates remarkable robustness. GenSIRR effectively

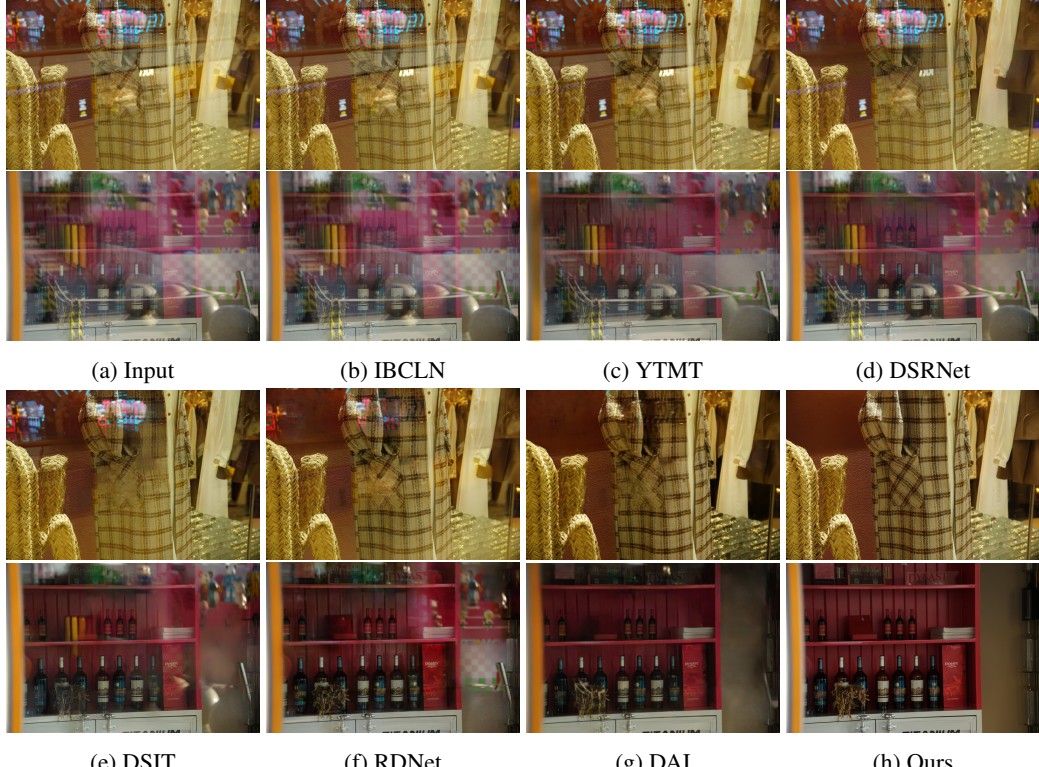

(a) Input      (b) IBCLN      (c) YTMT      (d) DSRNet

(e) DSIT      (f) RDNet      (g) DAI      (h) Ours

Figure 4: Qualitative comparisons on real-world cases. Please zoom in for more details.

removes complex reflections while preserving the natural appearance and fine details of the underlying scene, confirming its superior generalization capabilities.

**Downstream Applications.** The success of a reflection removal method is not just its visual quality, but its utility in improving the performance of downstream computer vision systems. Reflections can catastrophically degrade the performance of high-level vision models by introducing false objects, conflicting geometric cues, and semantic ambiguity. Here, we provide a qualitative comparison on monocular depth estimation and zero-shot image segmentation, as in Fig. 5. For each task, we compare the output of a state-of-the-art foundation model when run on: (1) the original image with reflections; (2) the output from previous SoTA, DAI, and DSIT; (3) the output from our method.

### 4.4 ABLATION ANALYSIS

To validate the effectiveness of the component in our GenSIRR framework, we conduct ablation studies on the challenging Real20 dataset. We analyze the contribution of the reflection-equivariant VAE and the learnable prompts. The quantitative results are summarized in Table 4 and Table 5.

**On the Rationale of the Reflection-Equivariant VAE.** Our hypothesis is that a structured latent space is helpful for SIRR. To verify this, we train a variant of our model where the reflection-equivariant VAE is replaced with a standard VAE trained only with a reconstruction loss ($\mathcal{L}_{recon}$), removing the linearity constraint ($\mathcal{L}_{equiv}$). As shown in Table 4, both changes lead to a significant drop in performance. This experiment confirms that the reflection-equivalence enables a precise and complete removal of the reflection.

**On the Efficacy of Learnable Prompts.** We argue that learnable prompts provide a more precise and effective form of task guidance than fixed text. To ablate this component, we replace our learnable prompts with a fixed text prompt, "Please remove the reflection in this image." The results in Table 5 show a marked degradation in performance. In some cases, it fails to remove complex reflection patterns. We also tried to randomly initialize the prompt or use RDNet (Zhao et al., 2025)'s prompt generator to generate a prompt. However, both choices failed to generate meaningful content, and the loss doesn't converge. A possible reason may involve the vast nature of the prompt

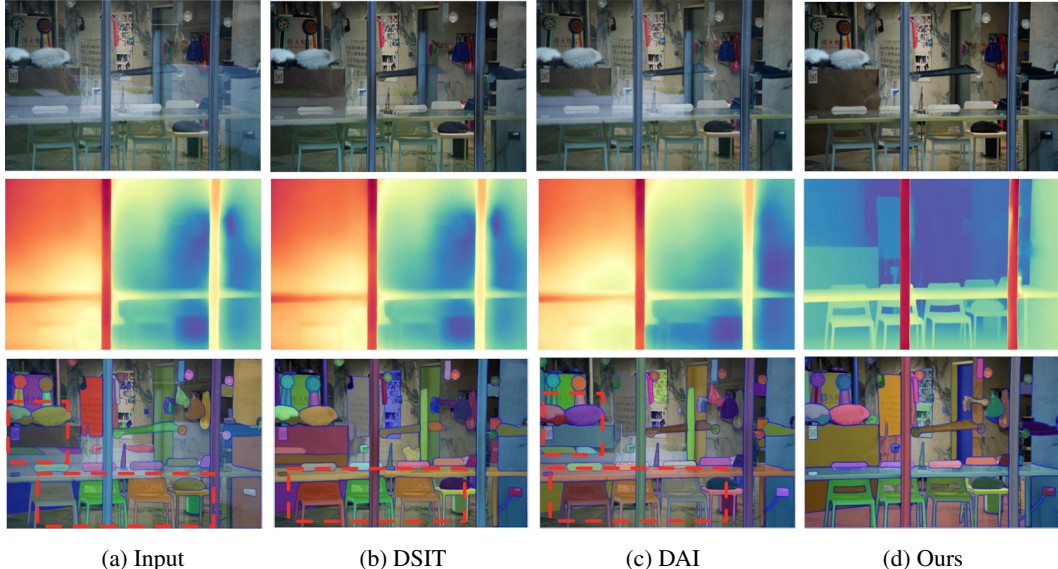

| (a) Input | (b) DSIT | (c) DAI | (d) Ours |

Figure 5: Qualitative evaluation on downstream vision tasks. *(Middle Rows)* Monocular depth estimation using Depth Anything V2 (Yang et al., 2024). Reflections in the original input induce severe geometric artifacts (e.g., false surfaces) in the predicted map. While the baseline method mitigates some errors, significant distortions remain. Our method yields a geometrically coherent depth map. *(Bottom Rows)* Zero-shot segmentation using the Segment Anything Model (Kirillov et al., 2023). Reflections pollute the semantic understanding of the scene. Our method eliminates these false positives, enabling SAM to correctly segment the scene content.

Table 4: The choice of training VAE.

| Choice | w/o Training | $\mathcal{L}_{\text{recon}}$ | Ours |
|---|---|---|---|
| PSNR | 25.72 | 25.79 | 27.27 |
| SSIM | 0.841 | 0.842 | 0.871 |

Table 5: The choice of prompt.

| Setup | Random | RDNet Prompt | Fix | Ours |
|---|---|---|---|---|
| PSNR | N/A | N/A | 26.52 | 27.27 |
| SSIM | N/A | N/A | 0.830 | 0.871 |

embedding space. A randomly initialized prompt lacks any semantic anchor, forcing the model to search an enormous space for a meaningful task vector from scratch. Without an initial direction, the optimization process fails to find a useful signal. Similarly, while the RDNet prompt generator is effective within its own architecture, its output is not semantically aligned with the pre-trained knowledge of model like FLUX.1, and thus also fails to provide a valid starting point.

## 5 CONCLUSION

In this work, we introduced GenSIRR, a novel framework that successfully addresses the long-standing challenges of single-image reflection removal, particularly in cases of heavy contamination and in-the-wild generalization. We argued that the limitations of prior methods stem from their discriminative nature, which restricts their ability to plausibly reconstruct occluded scene content. To overcome this, we proposed a paradigm shift: taming a pre-trained DiT-based image editing model to reframe SIRR as a guided generative task. Our core insight was to model the removal inside a structured latent space. We realized this through two designs: a reflection-equivariant VAE that extracts a compact latent prior of the unwanted reflection, and the learnable prompt token to learn precise task control. This strategy effectively transforms the powerful but unconstrained DiT into a precise and controllable restoration tool. Our extensive experiments validate the superiority of this approach, demonstrating that GenSIRR not only sets a new state-of-the-art on public benchmarks but, delivers exceptional visual quality and robustness on challenging real-world images. We believe GenSIRR presents a promising direction for leveraging large-scale generative models in complex image restoration tasks.

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

## A    LIMITATION

The primary limitation of GenSIRR is its inference speed. As a framework built upon an iterative diffusion process with a large transformer backbone, generating a single image requires multiple reverse sampling steps. This stands in stark contrast to previous CNN-based methods, which typically involve a single, fast forward pass. Our approach, therefore, prioritizes reconstruction quality, detail fidelity, and generalization over the computational efficiency required for real-time applications. However, this is an active area of research, and several promising avenues exist for future work to mitigate this issue. One direction is to explore model distillation, where the knowledge from our large GenSIRR model is transferred to a much smaller, faster student network. Additionally, recent advances in few-step or single-step sampling techniques, such as consistency models or rectified flow, could be adapted to our guided sampling scheme to reduce the number of required iterations without a significant drop in quality. Investigating these acceleration strategies will be a key step toward making our high-fidelity reflection removal practical for a broader range of applications.

## B    DISCLOSURE ON THE USAGE OF LLMS

Google's Gemini was used as a writing assistant in the preparation of this manuscript. Its use was limited to improving the grammar, clarity, and style of text drafted by the authors. All scientific ideas, experiments, and conclusions are the original work of the human authors, who take full responsibility for the paper's content.

## C    ADDITIONAL ANALYSIS

### C.1    ON THE POSSIBILITY OF LORA FINE-TUNING THE DIT

Our primary model is fully fine-tuned, a computationally intensive process. A natural question is whether a more parameter-efficient approach, such as Low-Rank Adaptation (LoRA) (Xu et al., 2024), could achieve comparable performance. To investigate this in a principled manner, we first analyzed the intrinsic rank of the task adaptation before attempting to train a LoRA-based model.

**Intrinsic Rank Analysis.** We computed the weight delta (the difference between our fully fine-tuned GenSIRR model and the pre-trained FLUX.1 base model) for each layer of the DiT. We then applied Principal Component Analysis (PCA) to this delta matrix to determine the number of principal components (i.e., the rank) required to explain 90% of the variance. A low intrinsic rank across layers would suggest that the task can be learned with a simple, low-rank update, making it an ideal candidate for LoRA. The results, visualized in Figure 6, were revealing. We found that the required rank was not uniformly low. Instead, we observed a distinct pattern where the intrinsic rank starts low, increases significantly in the middle layers of the network, often responsible for more abstract feature processing, and then drops again in the final output layers. Crucially, many of these internal blocks exhibited an intrinsic rank far exceeding 512, a value typically considered a high rank for LoRA. This analysis strongly suggests that SIRR is not a simple stylistic adaptation but requires substantial, high-rank modifications to the model's core internal representations.

### C.2    ON THE IMAGE SYNTHESIS ABILITY OF REFLECTION-EQUIVARIANT VAE

A critical consideration in our design is whether the proposed linearity constraint, while beneficial for our latent subtraction task, might inadvertently degrade the VAE's fundamental image synthesis capabilities. A VAE that cannot faithfully represent complex scenes would be a poor foundation for any restoration task. To investigate this, we evaluate the generative quality of our trained Reflection-Equivariant VAE against the baseline FLUX VAE from which it was adapted. We use the GenEval to quantitatively assess the model's ability to synthesize images based on a variety of compositional prompts. The benchmark evaluates performance across several attributes, including object presence, counting, and spatial relationships. As shown in Table 6, our Reflection-Equivariant VAE not only preserves but slightly *improves upon* the synthesis capabilities of the baseline model. We observe small but consistent performance gains across most categories, including single- and two-object scenes, color fidelity, and positional accuracy, leading to a higher overall score. It demonstrates that the linearity constraint does not introduce a detrimental trade-off. Instead, it appears to act as a

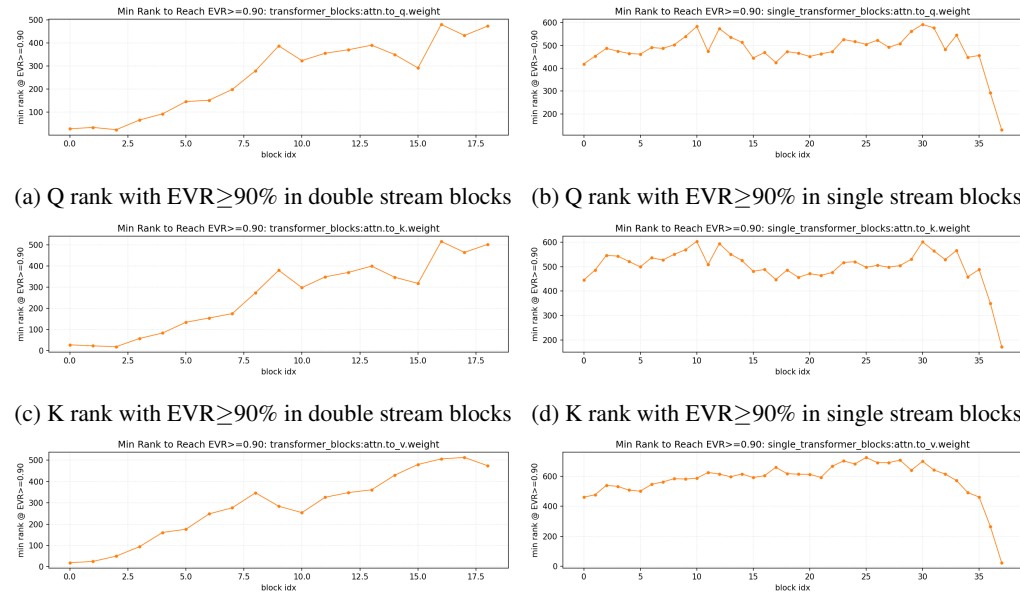

(a) Q rank with EVR≥90% in double stream blocks    (b) Q rank with EVR≥90% in single stream blocks

(c) K rank with EVR≥90% in double stream blocks    (d) K rank with EVR≥90% in single stream blocks

(e) V rank with EVR≥90% in double stream blocks    (f) V rank with EVR≥90% in single stream blocks

Figure 6: Intrinsic rank visualization for Q, K, and V projection matrices. We plot the minimum rank required to reach an Explained Variance Ratio (EVR) of 90% for the weight deltas before and after fine-tuning. The analysis is shown for both the double-stream (left column) and single-stream (right column) transformer blocks. The consistently high rank, especially in the middle layers, indicates that a low-rank update is insufficient for the SIRR task.

beneficial regularizer, encouraging the VAE to learn a more structured, disentangled, and semantically coherent latent space. This experiment validates our VAE design, confirming that we gain the geometric structure required for our task without sacrificing the model's core generative power.

Table 6: Comparison of image synthesis capabilities between our Reflection-Equivariant VAE and the baseline FLUX VAE, evaluated on the GenEval benchmark. Our VAE, trained with the additional linearity constraint, maintains and slightly improves upon the synthesis quality of the baseline.

| Compositional Attribute | Baseline VAE (FLUX) | Ours (Reflection-Equivariant) |
|---|---|---|
| Single Object | 97.81% | **99.38%** |
| Two Objects | 77.78% | **79.29%** |
| Counting | **72.81%** | 71.88% |
| Colors | 76.86% | **80.85%** |
| Position | 18.00% | **21.25%** |
| Color Attributes | 42.25% | **45.25%** |
| **Overall Score** | 0.64252 | **0.66316** |

# D    MORE VISUAL CASES

In this section, we bring more visual comparison on real-world cases in Fig. 7, Real20 in Fig. 8, SIR2 in Fig. 9, Nature and OpenRR in Fig. 10. As demonstrated in these cases, our method outperforms the previous methods consistently.

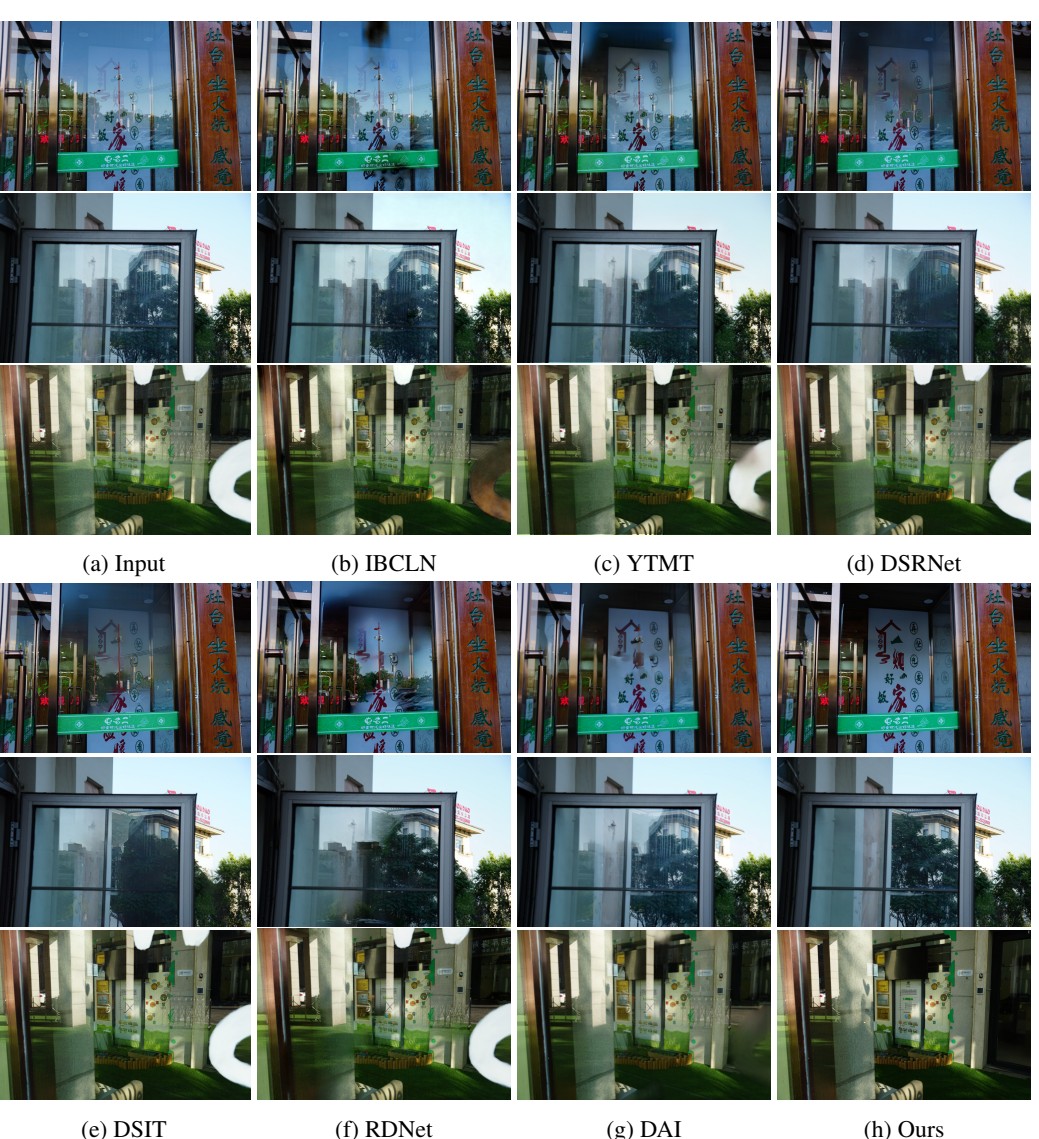

(a) Input      (b) IBCLN      (c) YTMT      (d) DSRNet

(e) DSIT      (f) RDNet      (g) DAI      (h) Ours

Figure 7: Qualitative comparisons on real-world cases. Please zoom in for more details.

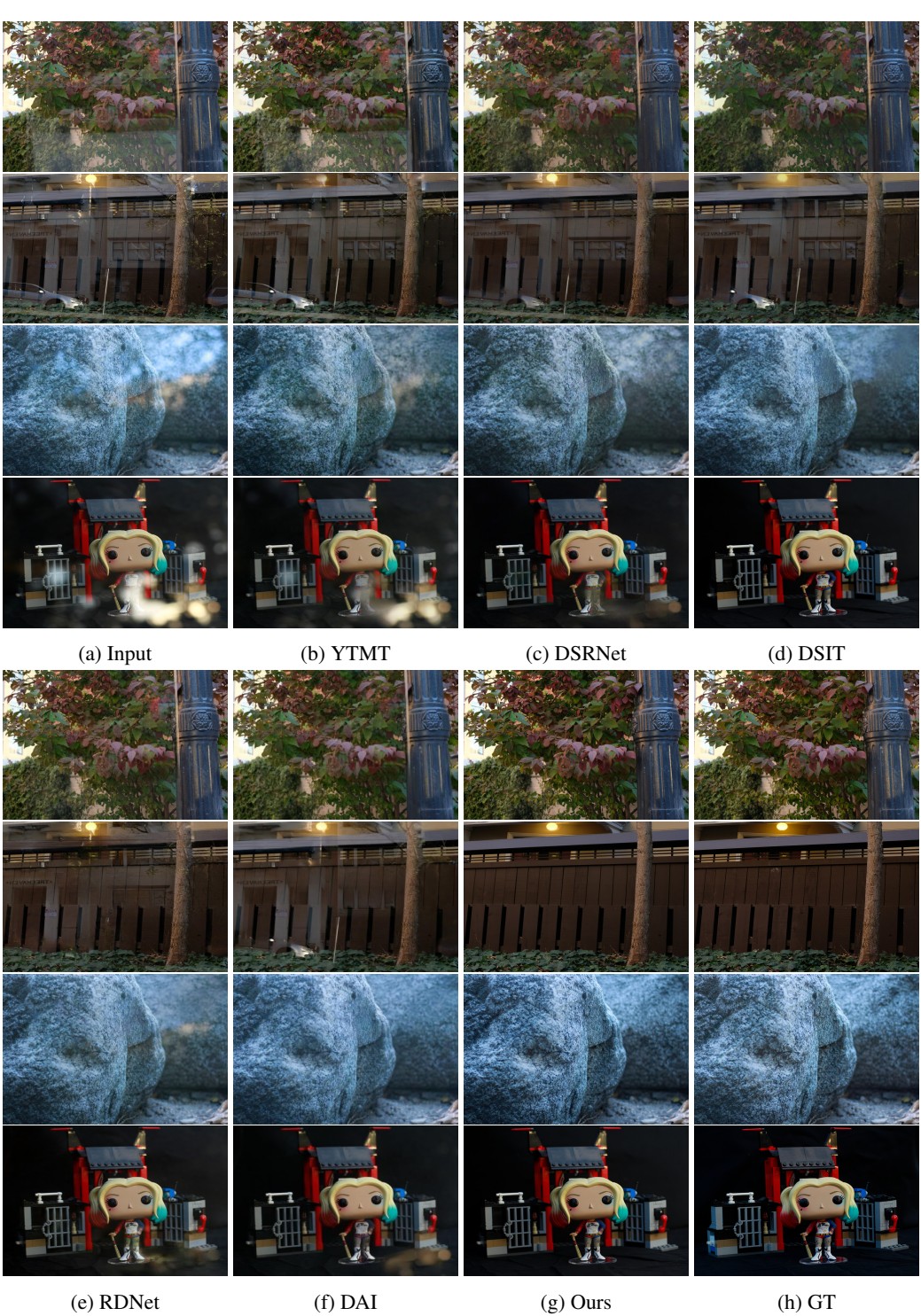

(a) Input      (b) YTMT      (c) DSRNet      (d) DSIT

(e) RDNet      (f) DAI      (g) Ours      (h) GT

Figure 8: Qualitative comparisons on Real20. Please zoom in for more details.

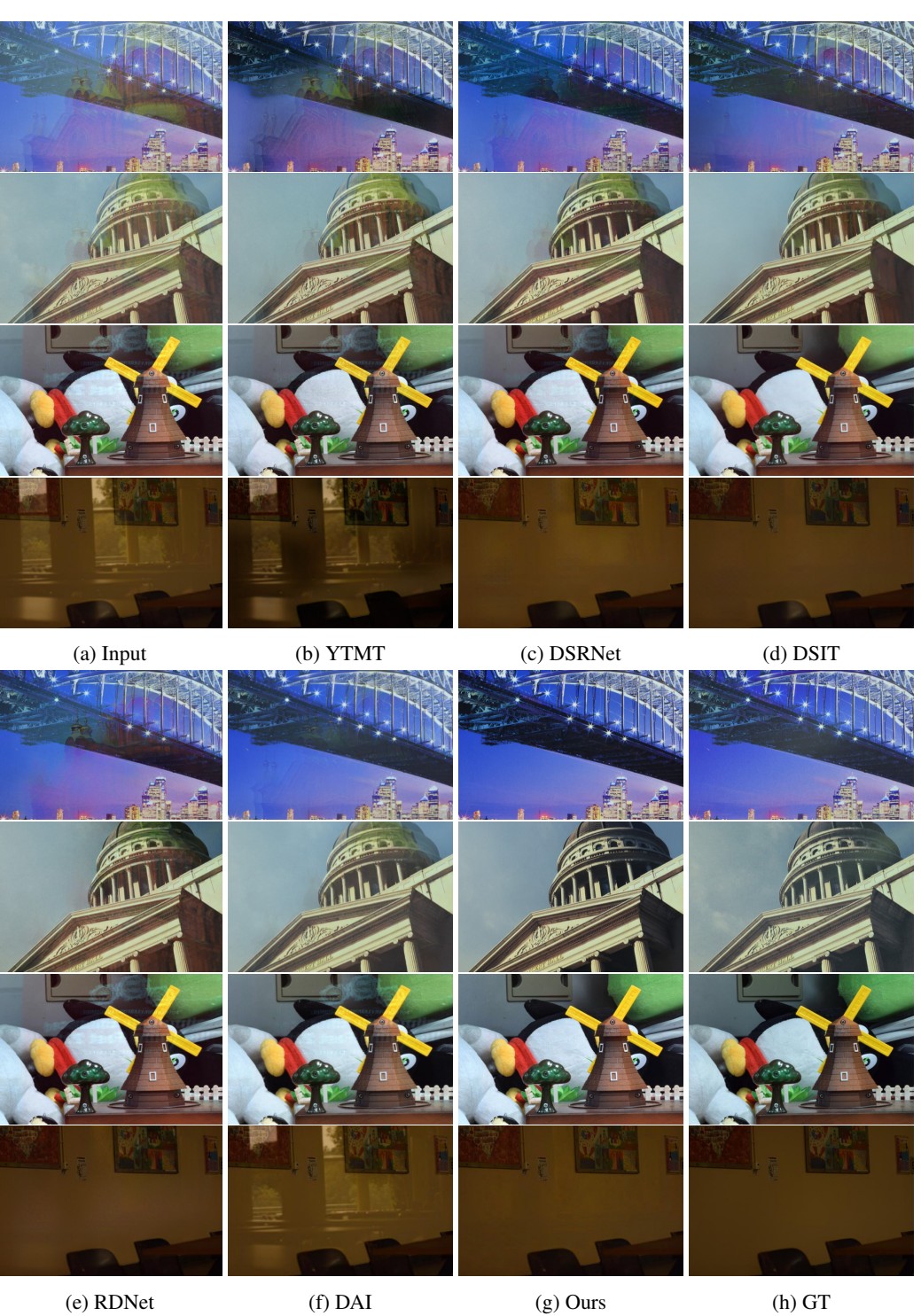

Figure 9: Qualitative comparisons on SIR2. Please zoom in for more details.

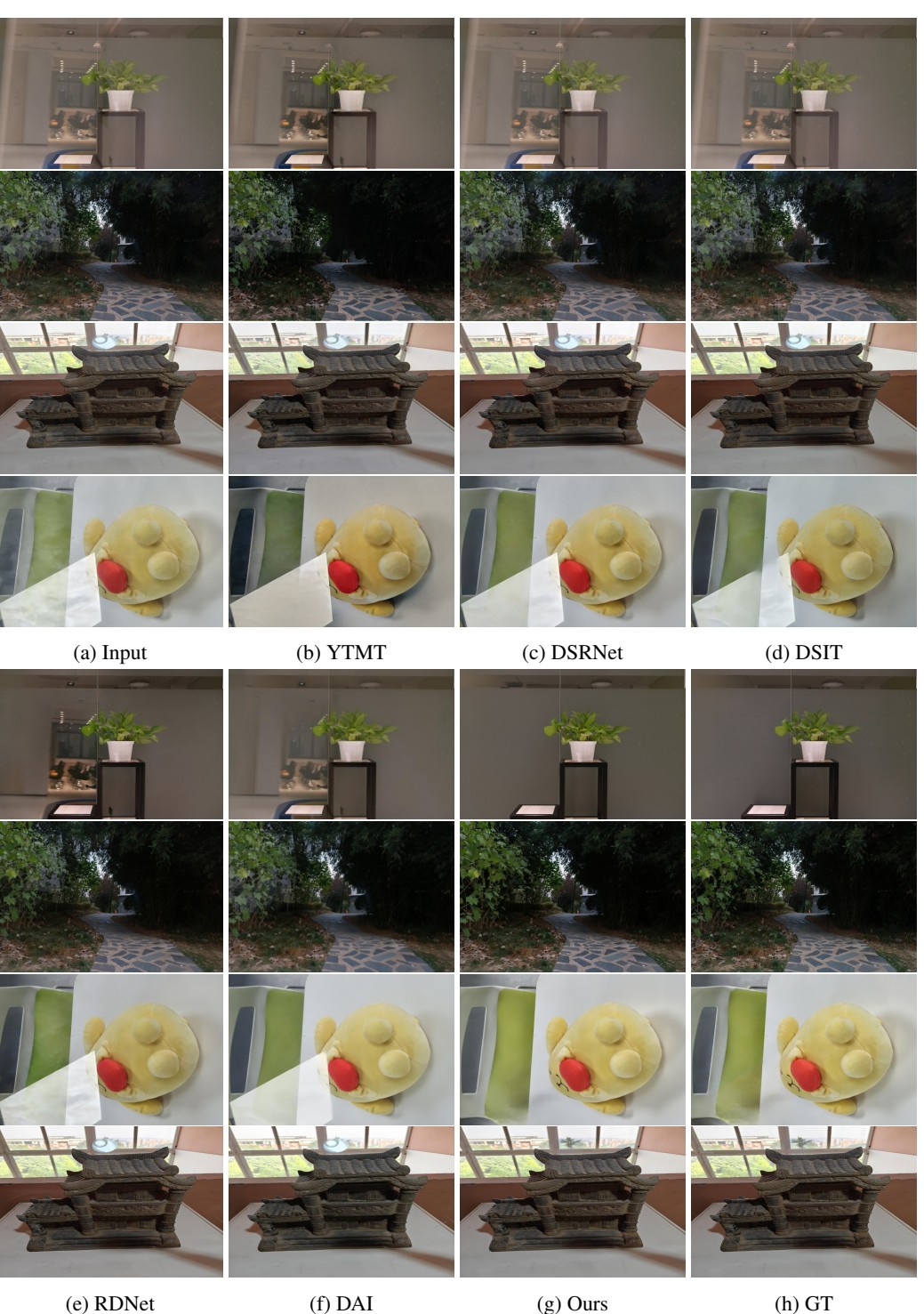

(a) Input      (b) YTMT      (c) DSRNet      (d) DSIT

(e) RDNet      (f) DAI      (g) Ours      (h) GT

Figure 10: Qualitative comparisons on Nature and OpenRR. Please zoom in for more details.

