# OpenReview forum: "From Generation to Restoration in Single-Image Reflection Removal"
_ICLR.cc/2026/Conference — ICLR 2026 Conference Withdrawn Submission_

### Official Review · Reviewer_xFt8 · 2025-10-31

**Soundness:** 2
**Presentation:** 2
**Contribution:** 2
**Rating:** 2
**Confidence:** 5

**Summary:**

This paper introduces a single-image reflection removal method. Its core idea is to adapt a pre-trained DiT into a restoration model. The author claims that such a design is able to transform a general-purpose image editing DiT into a precise and robust tool for reflection removal.

**Strengths:**

The authors propose some new thoughts on SIRR. The use of DiT can indeed solve some problems caused by the non-transmitted reflections.

**Weaknesses:**

1. At the beginning of the introduction, the authors mention, "When I cannot create, I do not understand". What is the correlation between this sentence and SIRR? For me, these opening remarks seem unnecessary and irrelevant.

2. The authors mention that the reflection may undermine both the aesthetic quality and the reliability of downstream vision tasks. It is beneficial to consider the needs of human vision and machine vision. However, the authors do not conduct any experiments to veify the impact of reflection on the machine vision. If the authors mention this in the introduction, it is better to conduct more experiments on whether their approach can improve the machine vision accuracy when the reflection is encountered.

3. It may be a new idea to use DiT in this place. However, using a generative model to estimate the missing information occluded by the non-transmitted reflections is not a new idea.  Before 2020, several methods were tried to use GAN for the estimation of missing information. Around 2021, some methods also propose incorporating information related to high-level tasks for further performance improvement. The authors need to do a more detailed survey to find the key difference between their proposed approach an the previous approach.

4.  It seems that quite a lot of restoration approaches are based on the DiT. It is better to clarify the differences.

**Questions:**

Please see my concerns in the weakness.

**Details Of Ethics Concerns:**

No ethics reviews are needed.

---

### Official Review · Reviewer_bJmM · 2025-10-31

**Soundness:** 3
**Presentation:** 3
**Contribution:** 3
**Rating:** 6
**Confidence:** 5

**Summary:**

This paper presents GenSIRR, a novel generative framework for single-image reflection removal (SIRR) that adapts a pre-trained Diffusion Transformer (DiT) into a precise restoration model. Unlike prior discriminative approaches, GenSIRR reframes reflection removal as a guided generation task through two main components: (1) a reflection-equivariant VAE that enforces a structured latent space aligned with the physical mixing of reflections, and (2) learnable task-specific prompts that replace ambiguous text instructions with optimized embeddings for precise control demonstrating strong generalization to real-world images.

**Strengths:**

• The paper introduces a well-motivated approach that reframes reflection removal as a guided generative task using a pre-trained Diffusion Transformer (DiT), effectively addressing the overfitting issue common in prior discriminative models.

 • The proposed reflection-equivariant VAE and learnable task-specific prompts are well-integrated and thoughtfully designed, enabling strong generalization to real-world reflection data.

 • The experimental results are consistently superior to current state-of-the-art methods by a clear margin, showing solid quantitative and qualitative improvements.

 • The experimental evaluation is extensive and diverse, covering multiple benchmark datasets, real-world “in-the-wild” cases, and even downstream tasks (depth estimation and segmentation), which further demonstrate the robustness and effectiveness of the proposed framework.

 • The paper is clearly written and well-organized.

**Weaknesses:**

The main weakness of the paper lies in its limited practical usability due to slow inference speed. Since the proposed method is built upon an iterative diffusion process with a large transformer backbone, it is computationally intensive and unsuitable for real-time applications such as mobile or interactive photography. While the authors acknowledge this limitation, the paper would be stronger if it included quantitative comparisons of inference time and computational cost against existing state-of-the-art reflection removal models (e.g., RDNet, DAI, DSIT). Such analysis would provide a clearer understanding of the trade-off between performance and efficiency.

·   	Another limitation concerns the use of an oversimplified physical model in the reconstruction loss. The paper defines reflection formation as , which is known to deviate from the real physical process of reflection formation and perform poorly on real-world data. This choice contrasts with the authors’ own use of a physically grounded reflection synthesis model (from RDNet) for generating synthetic training pairs. Employing the same physically grounded model in the loss formulation could have improved realism and alignment between the training objective and the actual image formation process.

**Questions:**

1-      Ablation Study Dataset Specification:
 The paper does not specify which dataset was used in the ablation experiments presented in Tables 4 and 5.
 while it can be inferred, it should be mentioned

2-      Choice of Reflection Formation Model:
 The reconstruction loss is defined using the simplified physical model , yet the authors mention that their synthetic data generation follows the physically grounded pipeline proposed by RDNet. Why was the loss not formulated based on this more accurate physical model instead of the simplified linear blend? Prior work has shown that the simplified model often yields weak generalization to real-world reflections, suggesting that a more faithful physical model could improve consistency and realism.

3-      Design of the Learnable Prompt Ablation:
 In the ablation study “On the Efficacy of Learnable Prompts”, the comparisons against random prompts and RDNet prompts are not particularly informative, as random prompts predictably fail to converge and RDNet prompts are inherently tailored to a different architecture. A more meaningful ablation would involve exploring different degrees of prompt learnability — for instance, partially learnable prompts (with frozen or restricted layers) versus fully learnable ones — to better illustrate how learnability influences performance and task adaptation.

4-      Efficiency Analysis:
 Since the paper acknowledges slow inference as a limitation, it would be valuable if the authors could quantify inference time and computational cost relative to existing reflection removal models. This would help contextualize the trade-off between reconstruction quality and practical deployment feasibility.

---

### Official Review · Reviewer_aCQQ · 2025-10-31

**Soundness:** 2
**Presentation:** 3
**Contribution:** 2
**Rating:** 2
**Confidence:** 5

**Summary:**

This paper propose a diffusion based model to tackle single image reflection removal task. The proposed method contains two modifications from a standard diffusion model. It firstly finetunes the VAE encoder part with LoRA to learn to minimize the distance between the input image embedding with the embedding of the mixture of reflection image and background image. Second, it also fine-tunes the output vector from text encoder to align with the diffusion model to optimize the prompts embeddings. Extensive experiments show that the proposed method outperforms the previous single image reflection removal baseline methods.

**Strengths:**

1. The proposed method outperforms the baseline methods quantitatively on a series of benchmarking datasets.
2. The qualitative examples show in Figure 3 are convincing.

**Weaknesses:**

1. What is the rational underneath the learnable prompts? Why does the trainable prompts help to clarify the ambiguous semantic meaning in text?  It is not clear how the vectors are optimized through the training of the DiT model if the text encoders are frozen.
2. Regarding the equivalence loss, it is necessary to provide more technical discussion and mathematical derivation to verify that the linear relationship remains in the embedding space. Eq.(2) is questionable.  Since the VAE encoder $E$ is not a linear function (due to its non-linearity layers), it is apparent that $E(I_{obs}) = E((1-\alpha)B + \alpha R)$  will not be equal to $(1-\alpha) E(B) + \alpha E(R)$ without any additional constraints.
3. What is the reason to use L1 loss in Eq. (3)?
4. In the abstract and introduction part, this paper is claimed to present a new framework. However, the framework is adopting the diffusion models (i.e. Flux) and it simply adopts a LoRA / text encoder fine-tuning.

**Questions:**

The authors are suggested to address the following two issues in the rebuttal period:
1. The proposed method does not match the claim and the title well.
2. There are some technical verification needed for the core contribution of the method.
3. The text prompts part needs more detailed clarification on the implementation.

---

### Official Review · Reviewer_irXS · 2025-11-01

**Soundness:** 2
**Presentation:** 3
**Contribution:** 2
**Rating:** 2
**Confidence:** 4

**Summary:**

This paper introduces a SIRR network based on the pre-trained DiT. Specifically, the VAE is fine-tuned using LoRA with a regularized latent space. Then, a set of learnable prompts are utilized during the fine-tuning of the pre-trained DiT network. Experiments show this method achieves SOTA performance.

**Strengths:**

- This paper is well-organized and presented with high quality.
- The motivation of learning a compact latent space specific to SIRR is clear.

**Weaknesses:**

- The key concern of this submission lies in its limited technical contribution. The idea of fine-tuning a reflection-equivariant VAE is originally introduced in this paper, but I cannot agree that the learnable prompts are technically novel.

- I remain conservative about the validity of Equation 2, though I believe this linear blend strategy works properly at the image level.

- It's unclear whether the training of Stage I is required for each dataset, or the fine-tuning of the VAE is one-time and can be extended to multiple datasets.

- Minor: I can't find the citation listed in line 197.

**Questions:**

- Q1: Can the idea of developing a reflection-equivariant latent space be applied to other image restoration tasks (*e.g.*, dehazing and low-light enhancement)?

- Please refer to the weakness part for other questions.

For the current submission, I can't give a score of 4, but I think the score of 2 is a bit low. Thus, my rating actually lies in the middle, and I'd like to disclose this information to the ACs, other reviewers, and the authors (rebuttals are welcome).

---

### Note · Authors · 2025-11-13

I have read and agree with the venue's withdrawal policy on behalf of myself and my co-authors.